# Acetylome Analyses Provide New Insights into the Effect of Chronic Intermittent Hypoxia on Hypothalamus-Dependent Endocrine Metabolism Impairment

**DOI:** 10.3390/biology13080559

**Published:** 2024-07-24

**Authors:** Yaru Kong, Jie Ji, Xiaojun Zhan, Weiheng Yan, Fan Liu, Pengfei Ye, Shan Wang, Jun Tai

**Affiliations:** 1Children’s Hospital Capital Institute of Paediatrics, Chinese Academy of Medical Sciences & Peking Union Medical College, Beijing 100020, China; kongyaru1995@163.com (Y.K.); ywh_tmu@126.com (W.Y.); liufan5023@163.com (F.L.); 2Graduate School of Peking Union Medical College, Beijing 100730, China; 3Department of Otolaryngology, Head and Neck Surgery, Children’s Hospital Capital Institute of Paediatrics, Beijing 100020, China; cnczhan81@163.com (X.Z.); yepengfei_doc@163.com (P.Y.); 4Department of Otolaryngology, Head and Neck Surgery, Beijing Children’s Hospital, Capital Medical University, National Center for Children’s Health, Beijing 100045, China; joyjee0819@163.com; 5Beijing Municipal Key Laboratory of Child Development and Nutriomics, Capital Institute of Paediatrics, Beijing 100020, China

**Keywords:** hypothalamus, chronic intermittent hypoxia, lysine acetylation, endocrine metabolism, paediatric obstructive sleep apnoea

## Abstract

**Simple Summary:**

The incidence of obstructive sleep apnea in children is increasing year by year, and its occurrence and related metabolic and endocrine lesions may be related to hypothalamic dysfunction. Lysine acetylation is a common post-translational modification that is essential for metabolism. The aim of this study was to determine the effects of chronic intermittent hypoxia-induced lysine acetylation on hypothalamic function in an infant mice model, simulating pediatric obstructive sleep apnoea patients. Our study revealed the lysine acetylome and proteomic profile of the hypothalamus induced by chronic intermittent hypoxia in juvenile mice. Differentially acetylated proteins were involved in endocrine metabolism, the citrate cycle (TCA cycle), synapse function, and circadian entrainment. Our findings suggest that chronic intermittent hypoxia may induce metabolic and endocrine dysregulation through modulation of hypothalamic lysine acetylation.

**Abstract:**

Paediatric obstructive sleep apnoea (OSA) is a highly prevalent sleep disorder resulting in chronic intermittent hypoxia (CIH) that has been linked to metabolism and endocrine impairment. Protein acetylation, which is a frequently occurring posttranslational modification, plays pivotal roles in the regulation of hypothalamic processes. However, the effects of CIH-induced global protein acetylation on hypothalamic function and endocrine metabolism remain poorly understood. To bridge this knowledge gap, we conducted a study utilizing liquid chromatography–mass spectrometry to analyse the lysine acetylome and proteome of the hypothalamus in healthy infantile mice exposed to 3 weeks of intermittent hypoxia (as a CIH model) compared to normoxic mice (as controls). Our analysis identified and quantified 2699 Kac sites in 2453 proteins. These acetylated proteins exhibited disruptions primarily in endocrine metabolism, the citrate cycle (TCA cycle), synapse function, and circadian entrainment. Additionally, we observed significant down-regulation of proteins that are known to be involved in endocrine hormone secretion. This study aimed to elucidate the molecular mechanisms underlying CIH-induced alterations in protein acetylation within the hypothalamus. By providing valuable insights into the pathophysiological processes associated with CIH and their impacts on hypothalamic function, our findings contribute to a deeper understanding of the consequences stemming from CIH-induced changes in protein acetylation within the hypothalamus as well as its potential role in endocrine impairment.

## 1. Introduction

Obstructive sleep apnoea (OSA) is the most common and severe sleep-disordered breathing (SDB) and is characterized by recurrent upper respiratory tract obstruction during sleep, thus resulting in chronic intermittent hypoxia (CIH) [1,2]. Its prevalence in the paediatric population may have gradually increased over the past decade in response to the increase in obesity and air pollution [3]. Paediatric OSA has an increased risk of growth retardation, cardiovascular diseases, neurocognitive disorders, and metabolic disorders [4,5,6]. Its pathogenesis is still unclear; moreover, as one of the brain regions that provides synaptic input to hypoglossal motor neurons, the hypothalamus may be related to upper airway collapse and OSA occurrence [7]. Furthermore, the multiple complications of OSA may be partially related to the activation of the stress system consisting of the sympathetic nervous system and the hypothalamic–pituitary–adrenal (HPA) axis [8,9,10].

The hypothalamus connects the endocrine system with other neural centres through complex neural circuits, thus making it a centre of neural integration for homeostatic functions, such as autonomic nervous function, neuroendocrine regulation, energy balance, sleep–wake cycles, and emotion processing [11]. Studies on the hypothalamic mechanism of OSA development have mainly focused on paraventricular nucleus (PVN) neurons, which strongly influence sympathetic/parasympathetic activity to drive hypertension [12,13,14,15]. In addition, the PVN projecting to the median ventricle plays a pivotal role in blood pressure regulation in CIH-induced hypertension [16,17]. In terms of neuroendocrine disorders, activation of the HPA axis and increased cortisol levels at night may be among the causes of neurocognitive dysfunction in OSA patients [18]. In addition, the hypothalamic growth-hormone-releasing hormone/growth hormone (GHRH/GH) axis is the basis of upper respiratory tract obstruction with growth retardation in children [19]. CIH exposure induces leptin resistance and changes in the expression of leptin-related proteins in the hypothalamic arcuate nucleus (ARC), thus leading to increased food intake and decreased conversion efficiency, thereby promoting weight gain [20,21]. Leptin administration may alleviate upper airway obstruction by activating the dorsomedial hypothalamus [22]. However, the hypothalamic mechanisms underlying the metabolic effects of CIH in the paediatric population have not been investigated.

Posttranslational modifications (PTMs) alter the properties of proteins by adding a modifying chemical group to an amino acid residue or another protein, thus making the proteome highly dynamic and functionally diverse, with modifications including phosphorylation, ubiquitination, acetylation, and methylation [23]. Lysine acetylation (Kac) is critical for metabolism and is the major PTM for almost all enzymes involved in glycolysis, gluconeogenesis, and the tricarboxylic acid cycle [24]. In addition, the effects of CIH on phenotypes resulting from Kac changes are only beginning to be investigated. CIH induces sympathetic excitation and blood pressure elevation in rats by inducing Kac of hypoxia-inducible factor (HIF)-1α [25,26]. Our previous studies demonstrated a comprehensive and detailed landscape of Kac in the hippocampus of CIH mice and linked this mechanism to CIH-induced cognitive dysfunction [27]. However, whether CIH alters the function of other brain regions such as the hypothalamus through Kac mechanisms is unknown.

In the present study, 3-week-old mice were subjected to CIH for 3 weeks to mimic children with OSA before adulthood. The CIH-induced global Kac changes in the hypothalamus and the possible functional abnormalities were then investigated. Proteomic analysis demonstrated that CIH caused global proteomic changes in the hypothalamus; specifically, it caused significant down-regulation of proteins known to be involved in endocrine hormone secretion. We also investigated global protein posttranslational Kac under CIH in the hypothalamus and identified key regulatory roles for Kac in endocrine metabolism, the citrate cycle (TCA cycle), and circadian entrainment. Our study aimed to elucidate the epigenetic mechanism of OSA-induced changes in brain function and explore possible treatments for patients with OSA complicated with metabolic disorders or other hypothalamus-related disorders.

## 2. Materials and Methods

### 2.1. Animals

3-week-old male BALB/c mice, weighing 9–11 g, were purchased from Sibeifu Biotechnology Co., Ltd. (Beijing, China). The mice were kept at a specific pathogen-free condition of 18–26 °C and 40–70% humidity, accompanied by a standard light–dark cycle. Experiments were performed in strict accordance with institutional ethical codes and were approved by Animal Care and Use Ethics Committee of the Capital Institute of Paediatrics (DWLL2021016).

### 2.2. CIH Protocol

Acclimatisation to the new environment took 3 days. Mice were randomly divided into control group and CIH group. Mice in the CIH group were exposed to CIH for 3 weeks in a hyperbaric oxygen chamber (Beijing Zhongshi Dichuang Technology Development Co., Ltd., Beijing, China) with alternating cycles of 1 min (oxygen concentration 5–21%) for 8 h/d (8:00 a.m.–4:00 p.m.) [17,28]. The control group was maintained in room air. At the end of the 3rd week of CIH, mice were euthanised and brains were rapidly saved for further studies.

### 2.3. Hematoxylin–Eosin (HE) Staining

Brain tissues were fixed with 4% paraformaldehyde solution for 48 h, washed thoroughly with phosphate buffered saline, dehydrated with alcohol, embedded in paraffin, and sectioned. They were stained with HE and then examined under a light microscope (Nikon Eclipse 80i, Nikon Corp., Tokyo, Japan).

### 2.4. Immunohistochemistry (IHC) Staining

Tissue antigens were repaired. Sections were baked, deparaffinised, dehydrated, and blocked with 5% goat serum. Then they were incubated with primary antibodies against acetyl-CoA synthetase (ACSS2, dilution 1:100, Abcam, Cambridge, UK), acyl-CoA dehydrogenase short chain (ACADS, dilution 1:100, Abcam), acyl-CoA oxidase (ACOX1, dilution 1:100, Abcam), or acetylated-Lysine (Kac, dilution 1:1000, Hangzhou Jingjie Biotechnology Co., Ltd., Hangzhou, China) overnight at 4 °C. After washing with PBS, sections were incubated for 1 h at room temperature using peroxiredox-conjugated secondary antibodies. They were observed under a light microscope (Nikon, 80i, Japan).

### 2.5. Protein Extraction and Trypsin Digestion

Total protein was extracted from the hypothalamus of control and CIH groups by lysis buffer (8 M urea, 1% protease inhibitor, 3 μM TSA, 50 mM NAM) and high-intensity ultrasonic processor (Scientz). After the supernatant containing the protein solution was obtained by centrifugation at 4 °C, the protein concentration was measured using the BCA kit (Beyotime Biotechnology, Shanghai, China) and the microplate reader. A final concentration of 20% trichloroacetic acid was added and centrifuged at 4 °C to obtain the deposition, washed with pre-cooled acetone, and dried. The final concentration of 200 mM tetraethylammonium bromide was added, followed by ultrasonic treatment. The trypsin was added at a ratio of 1:50 (protease: protein, m/m) and enzymolised overnight. Dithiothreitol with a final concentration of 5 mM was added and reduced at 56 °C for 30 min. Then the final concentration of 11 mM iodoacetamide was added and incubated at room temperature for 15 min away from light for alkylation. Proteomics and acetylomics used the same proteomic extraction and decomposition methods.

### 2.6. Acetylated Peptide Enrichment and High-Performance Liquid Chromatography (HPLC) Separation and Mass Spectrometry (MS)

HPLC-MS was performed using our previous method [27]. To be specific, the peptide was dissolved in immunoprecipitation (IP) buffer (100 mM NaCl, 1 mM EDTA, 50 mM Tris-HCl, 0.5% NP-40, pH 8.0). The supernatant was transferred to the antibody resin (PTM104, Jingjie), washed in advance, and shaken overnight at 4 °C. The resin was washed with IP buffer and deionised water. Then the resin-bound peptides were eluted with 0.1% trifluoroacetic acid and dried by vacuum freezing. The C18 ZipTips microchromatographic column was used for desalting.

The peptides were dissolved in mobile phase A by liquid chromatography and separated by Easy-nLC1000 ultra-high performance liquid phase system (Thermofisher, Waltham, MA, USA). Mobile phase A was an aqueous solution containing 0.1% formic acid and 2% acetonitrile. Mobile phase B was acetonitrile–aqueous solution containing 0.1% formic acid. The liquid phase gradient was 0–18 min, 9–24% B; 18–22 min, 24–35% B; 22–26 min, 35–90% B; 26–30 min, 90% B and the flow rate was 450 nl/min. The separated peptide segment was ionised using a Capillary ion source at a voltage of 1.6 kV. Data acquisition was performed using timsTOF Pro MS (Bruker, Billerica, MA, USA) and data-independent Parallel Cumulative Serial fragmentation (dia-PASEF) mode. The scanning range of primary mass spectrometry was 100–1700 *m*/*z*, and PASEF mode collection was performed 8 times after each primary mass spectrometry collection. The secondary mass spectrometry scan range was 425–1025 *m*/*z*, with one window every 25 *m*/*z*.

The raw data were processed using Spectronaut software (v. 17.0) and then compared with Mus_musculus_10090_SP_20230103.fasta to identify proteins. The Pfam database and PfamScan v.1.6 tool were used to identify and annotate protein domains. Protein subcellular localisation was annotated using WolF Psort (v. 1.0) software. The protein lysine modifications database (v. 3.0) [29] was used to analyse the association of PTMs in our data, identify novel acetylation sites, and explore the overlap with other PTM types. Pearson correlation coefficient (PCC) analysis was used to compare the intensity values of all samples. Differential expression of proteins and acetylated proteins was analysed using *p* value < 0.05 and foldchange >1.5 or <0.667 as criteria and further used for GO and KEGG analyses.

### 2.7. Statistical Analysis

All experiments were performed in triplicate and data were presented as mean ± standard deviation. Statistical analyses were performed with GraphPad Prism (v. 9.5). Group differences were assessed using Student’s *t*-test (for two groups).

## 3. Results

### 3.1. Identification of Numerous Proteins Exhibiting Differential Expression in the Proteome of the Hypothalamus under CIH

To investigate the impact of CIH on Kac changes in the hypothalamus, 3-week-old male C57BL/6J mice were subjected to a hyperbaric oxygen chamber for a period of 3 weeks to simulate CIH as they transitioned from adolescence to adulthood. To assess the impact of CIH on the hypothalamus at the molecular level, protein extractions from mouse hypothalamic tissue were conducted, followed by LC–MS analysis. We conducted three replications for each sample to ensure precision, and the PCC for Log_2_ LFQ intensity between replications was determined to be 1 (Figure 1A). Each identified protein was found to have at least one unique peptide. The total number of peptides and proteins that were identified after data filtering is displayed in Figure 1B. The length distribution of peptides that were identified by mass spectrometry met our stringent quality control criteria. In total, 7012 proteins were identified, 7002 of which were quantifiable. To identify differentially expressed proteins (DEPs), we established cut-offs for the foldchange (>1.5 or <0.667) and *p* value (<0.05). Proteins that did not meet these threshold parameters were excluded from further analysis. Among the DEPs, a total of 222 proteins exhibited significant modulation in the hypothalamus, with 103 up-regulated proteins and 119 down-regulated proteins (Figure 1C; Appendix A). Heatmap analysis demonstrated that CIH induced substantial alterations in the overall proteome profile within the hypothalamus (Figure 1D). To gain insights into their functional roles, we performed Clusters of Orthologous Groups (COGs) analysis on these DEPs. Our COGs analysis demonstrated that among these DEPs, a group consisting of 153 proteins was associated with information storage and processing, cellular process and signalling, and metabolism; moreover, it was poorly characterized. Another group comprising 21 proteins was linked to metabolic pathways such as energy production and conversion, inorganic ion transport/metabolism, carbohydrate/lipid transport and metabolism, and amino acid transport and metabolism. Furthermore, among the information processes involving 21 proteins, the most prominent included RNA processing and modification, chromatin structure and dynamics translation, ribosomal structure and biogenesis, transcription and replication, and recombination and repair (Figure 1E, Appendix A). To explore how CIH affects hypothalamic host functions, we conducted Gene Ontology (GO) enrichment analysis. This analysis demonstrated significant regulation of molecular functions (MFs) and cellular components (CCs), such as those depicted in Figure 1F,G (Appendix A), based on −Log_10_ (*p* values). The three most up-regulated CCs were synapses, Z discs, and actomyosin, whereas secretory granules, endoplasmic reticulum lumen and mitochondria were among the top down-regulated components (Figure 1F). As for MFs, the top three up-regulated functions included protein binding, catalytic activity, and dopamine binding, whereas peptidase inhibitor activity, enzyme binding, and sterol esterase activity were among the top down-regulated functions (Figure 1G). KEGG signalling pathway enrichment analysis of the identified proteins demonstrated that a majority of them are involved in divalent inorganic cation homeostasis, fatty acid metabolic process regulation, and protein tetramerization (Figure 1H and Appendix A). Additionally, significantly decreased expression levels of certain proteins enriched in internal secretion pathways (including endocrine hormone secretion, regulation of corticosteroid hormone secretion, and regulation of mineralocorticoid secretion) within the hypothalamus of CIH mice were found to be associated with more favourable prognostic outcomes (log-rank test, *p* < 0.05), as depicted in Figure 1I. In summary, these results showed that CIH caused global proteomic changes in the hypothalamus; specifically, it caused significant down-regulation of proteins known to be involved in endocrine hormone secretion.

### 3.2. Identification of Kac Proteins and Sites in the in the CIH–Hypothalamic Complex

Subsequently, histological analysis utilizing HE staining demonstrated that the hypothalamic cells were dispersed and that the nuclear macrocytoplasm was deeply stained, which identified damage within the hypothalamic cells, including cytolysis and cytoplasmic vacuolation (Figure 2A). Afterwards, we investigated whether there were any alterations in the levels of the enzymes Kac, ACSS2, ACADS, and ACOX1, which are involved in converting acetonyl-CoA within hypothalamic tissue. To accomplish this goal, we conducted immunohistochemical analysis on triplicate samples of hypothalamic tissues and compared them with normal tissues. Immunohistochemical analysis demonstrated that the levels of Kac and ACSS2 were increased, whereas the levels of ACADS and ACOX1 were decreased in the hypothalamic tissues of CIH mice compared with those of control mice (Figure 2B,C). These findings suggested that the up-regulation of ACSS2 may be relevant to the occurrence of CIH-induced hypothalamic damage in mice through the regulation of Kac levels. Despite CIH causing Kac and ACSS2 to increase, the specific mechanisms underlying the perturbation of the hypothalamic acetylome remain unclear. To obtain a whole landscape of the CIH-regulated acetylome, global acetylome analysis was performed in the hypothalamus of the control (*n* = 3) and CIH (*n* = 3) groups. HPLC-MS was applied to investigate Kac substrates in the CIH hypothalamus. *p* < 0.05 with foldchange > 1.5 or <0.667 was used to narrow down the list of potentially relevant changes in Kac levels. Principal component analysis (PCA) was used to distinguish between the control and CIH groups (Figure 3A). In total, 2453 proteins with 6165 acetyl sites were quantified (Figure 3B). Interestingly, 331 down-regulated Kac sites on 262 proteins and 1686 up-regulated Kac sites on 1055 proteins were detected in the hypothalamus of the CIH group (foldchange > 1.5 or <0.667; *p* < 0.05) compared with those in the Con group (Figure 3C,D and Appendix A). We identified 2699 acetylation sites distributed across 6610 acetylated proteins, accounting for 40.83% (Figure 3E). Among these Kac proteins, 1329 (49.2%) had a single Kac site, and 155 (5.7%) had more than six Kac sites (Figure 3F). Additionally, those associated with metabolism, such as Sptan1 (31 sites), Sptbn1 (25 sites), Aco2 (25 sites), Dync1h1 (25 sites), Dld (21 sites), Cnp (20 sites), Cltc (20 sites), Macf1 (18 sites), and Got2 (17 sites), exhibited the most abundant acetylation. It suggests that global changes in the hypothalamic lysine acetylome may contribute to alterations in endocrine metabolism. Although there is a deepening understanding of internal secretion metabolism dysfunction, there is still a lack of research on the mechanism by which Kac affects hypothalamic endocrine metabolism. Together, these results imply that CIH can cause global changes in protein acetylation in the hypothalamus.

### 3.3. Analysis of Characteristic Motifs at Acetylation Sites within the CIH–Hypothalamic Complex

To assess the amino acids surrounding the identified Kac sites, we employed iceLogo [30] against all of the mouse background sequences. Our findings demonstrated a greater occurrence of the negatively charged amino acid glutamic acid at positions −1 and +1 relative to the Kac sites (Figure 4A, Appendix A). The heatmap showed no significant difference between the distribution patterns of Kac and total protein lysine residues, suggesting the absence of a structural preference for Kac modifications (Figure 4B). Interestingly, Further analysis using Motif-X algorithms identified xxxxxxxxxx_K_Lxxxxxxxxx, xxxxxxxxxx_K_Hxxxxxxxxx, and xxxxxxxxxx_K_Ixxxxxxxxx as significantly overrepresented Kac site hotspots (Figure 4C and Appendix A). Additionally, NetSurfP analysis provided insights into the structures of proteins with Kac modifications. Our results showed that approximately 37.92% of the detected Kac sites were present in α-helices, whereas β-strands accounted for only 5.82%, and the remaining majority (67.56%) were located in coil structures (Figure 4D). There was no significant difference in surface accessibility between acetylated and non-acetylated proteins (Figure 4E). These findings highlight that Kac sites differentially modified by CIH in the hypothalamus can affect protein function by changing preferences for neighbouring amino acids and the secondary structure (but not surface accessibility).

### 3.4. Subcellular Distribution and GO Analysis of Significantly Up- and Down-Regulated Acetylated Proteins in CIH–Hypothalamic Tissue

Afterwards, we analysed the subcellular distribution of total acetylated proteins. There were differences in the subcellular locations of the up- and down-regulated acetylated proteins. As shown in Figure 5A, the up- and down-regulated differentially acetylated proteins mainly resided in the cytoplasm (Appendix A). Furthermore, a growing body of evidence has indicated that mitochondrial dysfunction is critical in the pathophysiology of cognitive impairment [31]. We found that 137 different acetylated proteins residing in mitochondria were up-regulated. Furthermore, our analysis demonstrated that Succinate-CoA ligase (Suclg1) can be acetylated at sites K66 and K81. Another example is Atp5f1c (oxidative phosphorylation, ATP synthase subunit gamma), for which we observed acetylation at the K208 site for the first time. However, 65 different acetylated proteins residing in mitochondria were down-regulated. Acetyl-CoA acetyltransferase (Acat1) can be acetylated at sites K63, K121, K187, K199, and K260. To gain insight into the biological functions and networks associated with differentially acetylated lysine residues, we performed GO analysis. The GO analysis of CCs demonstrated that up- and down-regulated differentially acetylated proteins were significantly enriched in the zona pellucida receptor complex and supramolecular complex, respectively (Figure 5B, Appendix A). The analysis of MFs demonstrated that up- and down-regulated acetylated proteins were significantly enriched in protein folding chaperone and microtubule minus-end binding, respectively (Figure 5B, Appendix A). These results indicate that CIH-induced up- and down-regulated acetylation was mainly involved in different MFs, including processing and cell components, in the cytoplasm and nucleus compartments.

### 3.5. Enrichment Clustering Protein Domain and Biological Function Analysis of the Kac Proteome in CIH–Hypothalamic Tissue

Subsequently, we categorized the significantly changed acetylated sites into four groups (Q1–Q4) based on foldchange (Figure 6A). GO analysis of biological progresses (BPs) demonstrated that Q1 was mainly involved in coenzyme A metabolic processes and coenzyme A biosynthetic processes, Q2 was mainly involved in pyridine nucleotide biosynthetic processes and nicotinamide nucleotide biosynthetic processes, Q3 was mainly involved in the regulation of sensory perception of sound and chemical synaptic transmission, and Q4 was mainly involved in the regulation of cell maturation and positive regulation of neuron differentiation (Figure 6B, Appendix A). Protein domain analysis demonstrated that Q1 Kac proteins were mainly involved in the PHD finger and anticodon binding domains; Q2 Kac proteins were mainly involved in the Malic enzyme, N-terminal domain, and NAD binding domain; Q3 Kac proteins were mainly involved in the FERM C-terminal PH-like domain and cation transporter/ATPase, N-terminus; and Q4 proteins were mainly involved in the immunoglobulin I-set domain and V-set domain (Figure 6C, Appendix A). The WikiPathways analysis revealed that Q1 Kac proteins were mainly involved in fatty acid beta-oxidation and biosynthesis, Q2 Kac proteins were mainly involved in the TCA cycle and mRNA processing, Q3 Kac proteins were mainly involved in calcium regulation in cardiac cells and the Mapk signalling pathway, and Q4 Kac proteins were mainly involved in cytoplasmic ribosomal proteins (Figure 6D, Appendix A). These results indicate that CIH affects multiple biological functions and protein domains by modulating the acetylation levels of relevant proteins in the hypothalamus in CIH mice.

### 3.6. Effects of Acetylated Lysine in CIH on Key Hypothalamic Cellular Metabolism and Synapse Functions

To learn more about the functions of the acetylated proteins, we performed KEGG signalling pathway analysis. For the up-regulated sites, the most endocrine functions shared by these acetylated proteins were mainly related to endocrine and other factor-regulated calcium reabsorption, thyroid hormone synthesis, and insulin secretion (Figure 7A). It also included synapse functions which were mainly related to Glutamatergic synapse, GABAergic synapses, and circadian entrainment. Proteins involved in the insulin secretion pathway which acetylation was up-regulated in CIH included 55 K and 57 K in Stx1a; 95 K, 161 K, 168 K, 404 K, 410 K, and 455 K in HSP90b1 within the thyroid hormone synthesis pathway; Slc1a2 (29 K, 40 K, and 569 K) and Slc1a3 (42 K, 118 K, 152 K, and 191 K) involved in glutamatergic synapse pathway; K250 in the Camk2a; and 603 K in Ryr2 within the circadian entrainment pathway. For the down-regulated sites, the most enriched cellular metabolism pathways shared by these acetylated proteins were mainly related to the citrate cycle (TCA cycle), pyruvate metabolism, and glycolysis/gluconeogenesis (Figure 7B, Appendix A). Proteins involved in the citrate cycle (TCA cycle) in which acetylation was down-regulated in CIH included dihydrolipoyl dehydrogenase (Dld K346), isocitrate dehydrogenase (Idh1 K243, Idh2 K193, and Idh3g K226), pyruvate dehydrogenase (Pdha1 K313), ATP-citrate synthase (Acly K544), aconitate hydratase (Aco2 K573), succinate dehydrogenase (Sdha K250), dihydrolipoyllysine-residue acetyltransferase component of pyruvate dehydrogenase complex (Dlat1 K468), and acetyl-CoA acetyltransferase (Acat1 K63, K121, K187, K199, and K260). Proteins involved in glycolysis/gluconeogenesis included pyruvate kinase (Pkm K433), ATP-dependent 6-phosphofructokinase (Pfkl K726), and glyceraldehyde-3-phosphate dehydrogenase (Gapdh K192 and K123). The down-regulated crotonylation sites of glycolysis/gluconeogenesis regulators included a diverse set of proteins, including HK2 (HK2 478cr) and PGK1 (PGK1 199cr) (Figure 7B). Furthermore, a key function of histone modification is the epigenetic regulation of gene activation and silencing. Images of the histones and histone variants with acetylation are shown in Appendix A and Appendix A. We identified and quantified a total of 20 acetylation core histone lysine sites, with 14 showing higher levels of acetylation in the hypothalamus of mice exposed to CIH. For instance, we observed a 2.7-fold increase in H1.0K55Ac and a 1.5-fold increase in H4K78Ac in the hypothalami of mice in the CIH group. There were minimal changes in Kac levels at all of the other sites. These findings further support the idea that CIH can positively regulate hypothalamic histone acetylation in the hypothalamus. We also detected histone acetylation on classical and variant histones, including modifications on H1K55, H1K59, H1.4K63, 75, 58, 90, 97, 106, and 168; H2BK6, K44, and K47; and H4K60 and K78, which play crucial roles in epigenetic regulation. Acetylation sites associated with transcriptional activation include H3K122ac and H4K79ac. These double- or higher-order modifications suggest a more intricate level of gene regulation through tandem acetylation epigenetic mechanisms. Our study demonstrated the pivotal regulatory role of Kac in endocrine hormone and synapse regulation and the citric acid cycle (TCA cycle) metabolism in the hypothalamus under CIH.

## 4. Discussion

OSA is a prevalent SDB characterised by snoring, obstruction of the upper respiratory tract during sleep, intermittent CIH, and sleep fragmentation, thus leading to significant health implications [32]. OSA affects at least 2% of adolescents throughout the world [33]. CIH has been shown to impact cognitive functions and also lead to metabolic impairment. However, the precise mechanisms are not fully understood. The hypothalamus, which is a crucial component of the brain, plays a fundamental role in controlling energy balance in vertebrates [34]. Previous studies have indicated that the epigenetic pattern of the hypothalamus changes during animal development and other physiological processes. Among these, epigenetic changes, including protein modification and DNA methylation, play a critical role in regulating neurogenesis, cognition, neurodegeneration, and other brain functions [35,36,37]. This scenario is particularly true for the hypothalamus, as programmed epigenetic changes have been observed during animal development. An important mechanism for modulating gene–environment interactions is the acetylation of lysine residues in histone and nonhistone proteins. The deacetylation of STAT3 mediated by HDAC5 and Nur77 is an important regulator of hypothalamic leptin signalling and ensures the appropriate control of energy homeostasis within the central nervous system (CNS) [38,39,40,41]. We have recently reported the effect of altered hippocampal protein acetylation on cognitive function in CIH mice [27]. In the existing modelling and protein acetylation CIH group learning context of this study, in order to simulate paediatric patients, 3-week-old mice were used and the corresponding CIH time was reduced from 6 weeks to 3 weeks. Our aim was to investigate the changes in the proteome and acetylome profiles of the hypothalamus in CIH mice.

Proteome profiling demonstrated that a total of 222 proteins were significantly differentially modulated between the CIH–hypothalamus group and the control group (Figure 1B). There were significantly decreased expression levels of certain proteins enriched in internal secretion pathways, including endocrine hormone secretion, regulation of corticosteroid hormone secretion, and regulation of mineralocorticoid secretion, within the hypothalamus of the CIH mice (Figure 1I). We identified 2453 proteins with 6165 acetyl sites in the hypothalamus of the CIH mice (Figure 3B). Further analysis of the quantified sites and proteins revealed that 331 down-regulated Kac sites on 262 proteins and 1686 up-regulated Kac sites on 1055 proteins (foldchange > 1.5 or <0.667; *p* < 0.05) were differentially regulated by CIH exposure. The acetylated lysine regions generated 33 conserved motifs (Appendix A). Consistently, Motif-X algorithms analysis identified xxxxxxxxxx_K_Lxxxxxxxxx, xxxxxxxxxx_K_Hxxxxxxxxx, and xxxxxxxxxx_K_Ixxxxxxxxx as significantly overrepresented Kac site hotspots (Figure 4C). Kac plays key regulatory roles in endocrine hormone and synapse regulation and the citric acid cycle (TCA cycle) metabolism (Figure 7A,B). Accordingly, acetylation may be important in the development of the hypothalamus and is associated with endocrine dysfunction under CIH exposure.

Our results suggest that CIH exposure significantly changed the acetylome profile of the hypothalamus and may be associated with the pathogenesis of related endocrine impairments. Our investigation showed that the up-regulated pathways were most enriched in endocrine secretion related to thyroid hormone synthesis, as well as endocrine regulation of calcium reabsorption, pancreatic secretion, insulin secretion, and salivary secretion. Pathways associated with proteins exhibiting the most enrichment were related to endocrine and metabolic disease (Figure 7A). We detected acetylation of 55 K and 57 K in Stx1a within the insulin secretion pathway. In type 2 diabetes (T2D), severely reduced islet Stx1a levels contribute to insulin secretory deficiency. Stx1a-KO mice exhibited significant defects in insulin exocytosis, thus indicating the potential impact of global Stx1a deletion on neuronal and endocrine secretions. Recent reports have also demonstrated that disturbances in the HPA axis affect the release of corticosterone and catecholamine, both of which play crucial roles in glucose homeostasis [42,43,44]. We also detected acetylation of 95 K, 161 K, 168 K, 404 K, 410 K, and 455 K in HSP90b1 within the thyroid hormone synthesis pathway. Thyroid hormones (THs) exert a profound influence on vertebrate development and play a pivotal role in facilitating metabolic responses to environmental stimuli, such as temperature fluctuations, stress, inflammation, and nutrient deficiencies [45]. Deacetylase HDACs play a role in the regulation of thyroid hormone pathways within the hypothalamus. Specifically, it has been proposed that HDAC4 and HDAC6 may serve as key components linking epigenetic modulation of hormone pathways in the hypothalamus [46,47]. It is important to note that HDAC expression serves as an epigenetic switch that can have widespread effects on these pathways by repressing thyroid hormones. The repressive effect of HDACs on gene expression has been shown to significantly impact THR-induced transcription, particularly through their influence on target genes such as Hsp90 [48]. Our study suggested that acetylation of Hsp90 is involved in thyroid hormone function in the hypothalamus under CIH exposure. Hsp90 is subjected to various PTMs, such as phosphorylation and acetylation [49]. These studies examined the role of Hsp90 acetylation in mediating glucocerebrosidase activity in Gaucher disease and tau phosphorylation in Alzheimer’s disease. Therefore, our findings provide novel insights into the underlying mechanisms of CIH-induced endocrine and metabolic impairment and suggest that proper regulation and maintenance of thyroid hormone function is important in the maintenance of endocrine health.

The aetiology of metabolic syndrome has been linked to an increase in sympathetic nerve activity and the HPA axis [50]. This physiological state, with its antagonistic effects on insulin actions, serves to prevent hypoglycaemia while also disrupting energy balance by redirecting energy fluxes from muscles towards abdominal fat depots. The synergistic interplay between hyperinsulinemia and hypercortisolism contributes to the promotion of abdominal visceral obesity and insulin resistance, which are fundamental pathophysiological manifestations of metabolic syndrome [51,52]. It has been postulated that the activation of the HPA axis induced by hyperinsulinemia promotes the occurrence and development of metabolic syndrome and its associated consequences [53,54].

Our analysis also demonstrated that the up-regulated pathways associated with proteins exhibiting the most enrichment were related to neurodegenerative disease (Appendix A) and involved in synapse functions related to GABAergic synapses, the synaptic vesicle cycle, and glutamatergic synapses. Slc1a2 can genuinely ameliorate epilepsy through the glutamatergic synapse pathway, mitigate neuronal loss, and suppress astrocytosis and inflammatory responses [55]. Furthermore, our study demonstrated that upon exposure to CIH, Slc1a3 and Slc1a2, which are crucial glutamate transporters involved in glutamatergic synapses, undergo acetylation at many lysine residues, such as Slc1a2 (29 K, 40 K, and 569 K) and Slc1a3 (42 K, 118 K, 152 K, and 191 K). Our findings align with existing evidence that acetylation activates glutamate transporters. At glutamatergic synapses, astrocytes play a crucial role in the regulation of neural activity by actively removing the byproducts of synaptic transmission, such as potassium, glutamate, and excess water. These substances are then efficiently redistributed throughout the intricate glial network, where they can be either recycled for neuronal use or released into the vascular compartment [56,57]. The maintenance of optimal levels of these molecules within the synaptic cleft is facilitated by specialized glutamate transporters SLC1A2 and SLC1A3 that operate within a highly integrated cell network known as a syncytium. This intricate system effectively restores the balance of synaptic activity by regulating and replenishing essential neurotransmitter and ion levels to their basal state.

In addition, we also detected acetylation of K250 in the Camk2a protein and 603 K in the Ryr2 protein within the circadian entrainment pathway. In mammals, nocturnal light information is transmitted to the circadian clock in the suprachiasmatic nucleus of the hypothalamus through glutamate released from retinal projections [58]. Clock resetting requires the activation of ionotropic glutamate receptors, which mediate Ca^2+^ influx. The depletion of intracellular Ca^2+^ stores during the early night has been shown to block the effects of glutamate. Activators of ryanodine receptors induce phase resetting only at the early stages of night; moreover, inhibitors selectively block delays induced by light and glutamate [59]. This finding suggested that the release of intracellular Ca^2+^ through ryanodine receptors is involved in the light-induced phase delay of the circadian clock restricted to the early phase of night. PTMs at specific sites in Camk2a have been demonstrated to modulate neuronal plasticity [60]. Additionally, phosphorylation of the T286 residue in Camk2a is essential for neuronal function and development. The RyR1 (type 1 ryanodine receptor) is widely expressed in the brain, with high levels found in the cerebellum, hippocampus, and hypothalamus [61]. The RyR1 plays a role in the voltage-induced calcium release (VICaR) mechanism within hypothalamic cells [62]. Taken together, acetylation of both the Camk2a and Ryr2 proteins may be implicated in aspects of circadian entrainment associated with hypothalamus function under CIH in the future.

The down-regulated pathways were enriched in cellular metabolism pathways, including the citrate cycle (TCA cycle), pyruvate metabolism, and glycolysis/gluconeogenesis (Figure 7B). Additionally, our study demonstrated that the protein domains enriched in the differentially abundant proteins were primarily associated with the immunoglobulin domain and EF-hand domain pair domains (Figure 6C). It indicated that hypothalamic dysfunction may be related to decreased glucose metabolism. Consistent with the findings of a previous study, we observed decreases in the acetylation of Dld on K346; Idh1 on K243; Idh2 on K193; Idh3g on K226; Pdha1 on K313; Acly on K544; Aco2 on K573; Sdha on K250; Dlat1 on K468; and Acat1 on K63, K121, K187, K199, and K260 in the TCA cycle, as well as Pkm on K433, Pfkl on K726, and Gapdh on K192 and K123 in glycolysis/gluconeogenesis following CIH exposure. This metabolic switch may exacerbate CIH-induced glycolytic and TCA enzyme acetylation, thus further impairing hypothalamic function.

## 5. Conclusions

In conclusion, our study aimed to elucidate the molecular mechanisms underlying hypothalamic endocrine dysfunction induced by CIH. In pursuit of this goal, we conducted lysine acetylome profiling to construct a comprehensive and intricate map of Kac in the hypothalamus. Our analysis revealed 2453 proteins with 6165 acetylation sites, with the identification of 1317 differentially regulated acetylation sites on 2017 proteins in the hypothalamus under CIH. Our findings suggest that dysregulation of endocrine secretion, synaptic functions, circadian entrainment, and the citrate cycle (TCA cycle) may contribute to exacerbating endocrine impairment after CIH. Overall, the elucidation of the mechanisms governing CIH-induced alterations in hypothalamic acetylation has the potential to inform the development of targeted interventions for mitigating endocrine metabolic decline associated with CIH.

## Figures and Tables

**Figure 1 biology-13-00559-f001:**
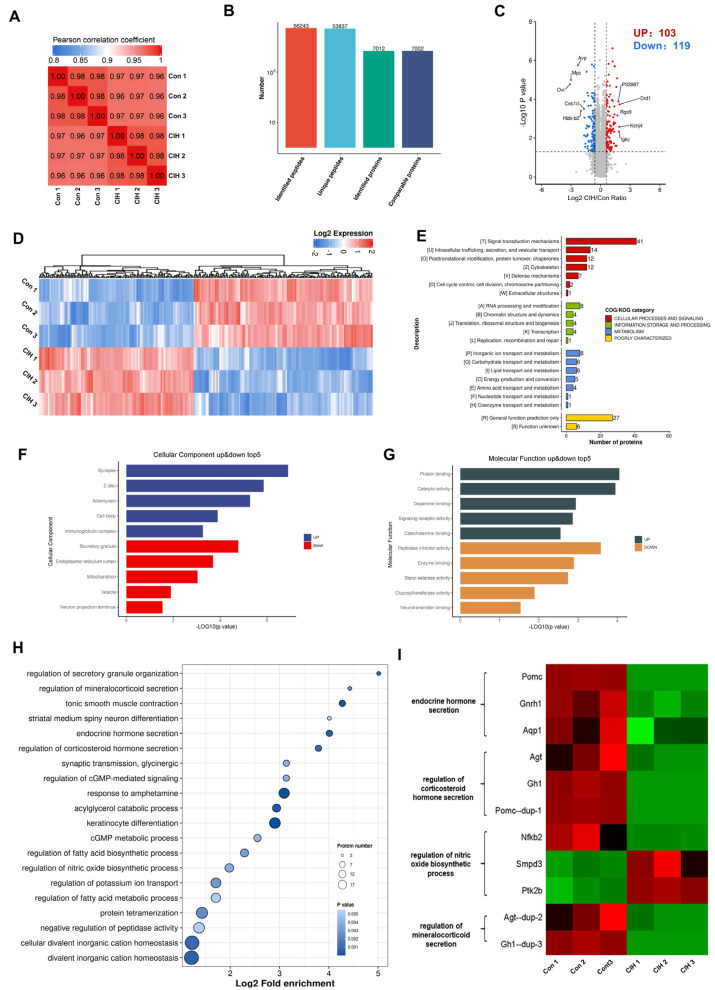
Proteomic changes in the hypothalamus of CIH mice. Mice in the CIH group were exposed to CIH for 3 weeks. At the end of the third week, control and CIH mice were sacrificed and the hypothalamus was preserved. Each group of three hypothalami is used for proteomics and acetylomics. (**A**) PCC analysis was used to compare the intensity values of all samples. (**B**) The total number of peptides and proteins that were identified after data filtering. (**C**) A volcano plot and (**D**) hierarchical clustering heatmap present DEPs, which were screened according to foldchange >1.5 or <0.667, with *p* value < 0.5. (**E**) COG, (**F**) GO-CC, (**G**) GO-MF, and (**H**) KEGG analyses were performed on DEPs. (**I**) The heatmap shows the protein expression in endocrine-related pathways. CIH: chronic intermittent hypoxia; PCC: Pearson correlation coefficient. DEPs: differentially expressed proteins; COGs: Clusters of Orthologous Groups; GO: Gene Ontology; CC: Cellular Component; MF: Molecular Function; KEGG: Kyoto Encyclopaedia of Genes and Genomes.

**Figure 2 biology-13-00559-f002:**
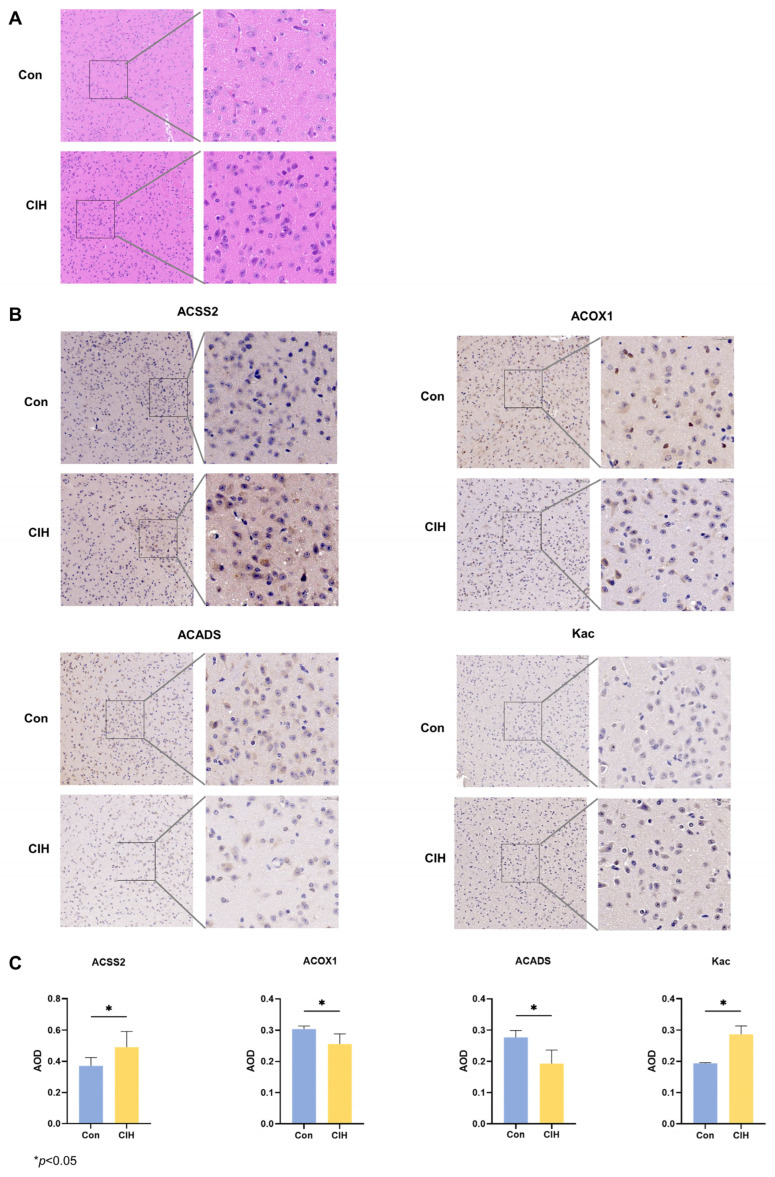
Expression changes of acetylation-related enzymes in the hypothalamus of CIH mice. (**A**) HE staining. (**B**,**C**) IHC staining was used to detect the protein expression of ACSS2, ACADS, ACOX1, and Kac. All experiments were repeated three times. All data are presented as the mean ± standard deviation. CIH: chronic intermittent hypoxia; IHC: immunohistochemistry; ACSS2: acetyl-CoA synthetase; ACADS: acyl-CoA dehydrogenase short chain; ACOX1: acyl-CoA oxidase; Kac: acetylated Lysine.

**Figure 3 biology-13-00559-f003:**
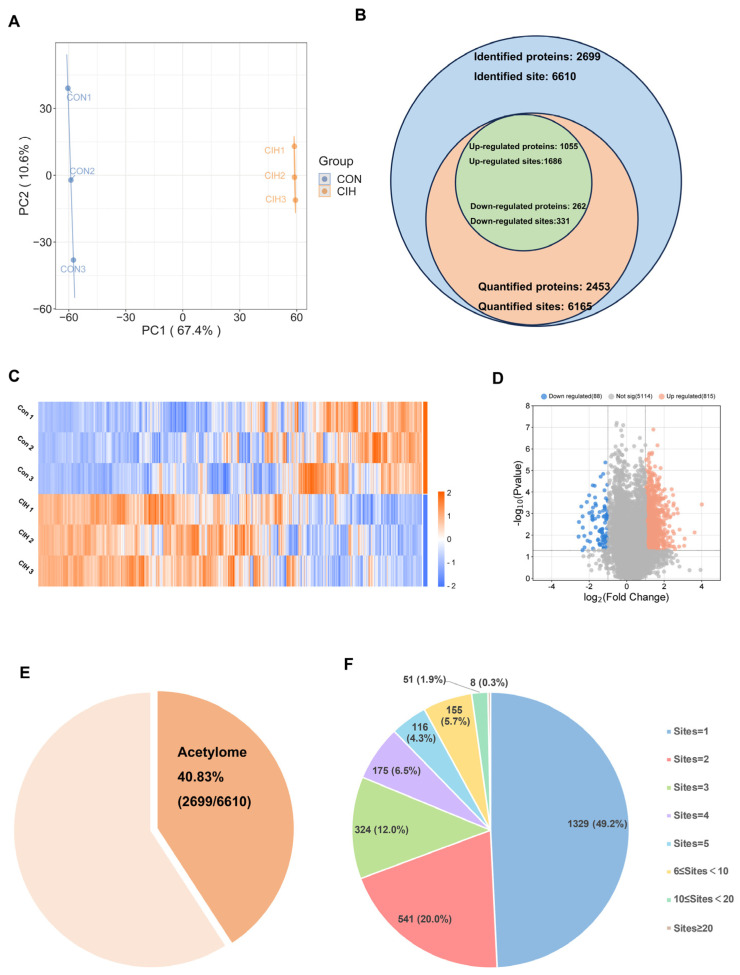
Characteristic motif analysis of acetylation sites in the hypothalamus of CIH mice. (**A**) PCA was used to characterize the signatures of the control and CIH groups and distinguish them. (**B**) Numbers of identified and quantified proteins and Kac sites. (**C**)The hierarchical clustering heatmap and (**D**) volcano plot present differential Kac sites, which were screened according to foldchange > 1.5 or <0.667, with *p* value < 0.5. (**E**) The proportion of Kac proteins to total identified proteins; (**F**) the number of Kac sites in Kac proteins. CIH: chronic intermittent hypoxia; PCA: principal component analysis; Kac: Lysine acetylation.

**Figure 4 biology-13-00559-f004:**
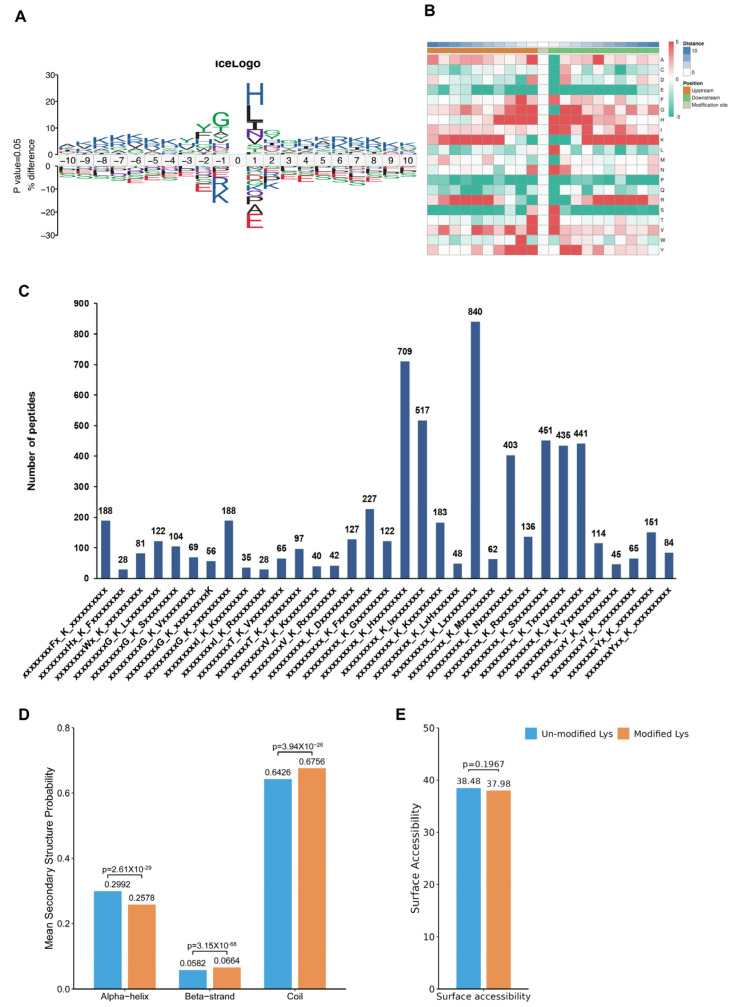
Analysis of molecular characteristic groups of acetylation sites. (**A**) The iceLogo method shows the frequency changes of amino acids near acetylation sites; (**B**) the heatmap showed the structural preference of all acetylation sites; (**C**) statistics of motif changes at acetylation sites; (**D**) statistics of the α-helix, β-strand, and coil of the identified Kac sites. (**E**) Statistics of the surface accessibility of the identified Kac sites. Kac: Lysine acetylation.

**Figure 5 biology-13-00559-f005:**
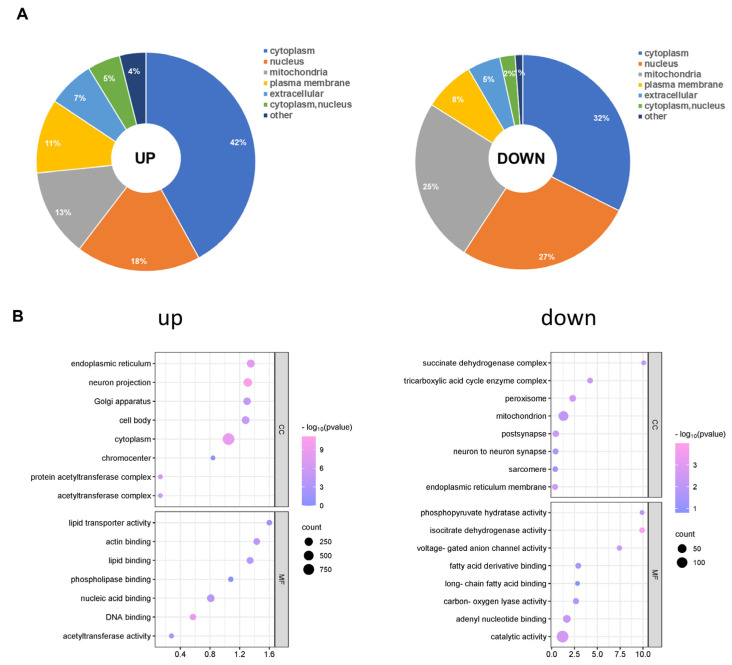
Subcellular distribution and GO analysis of acetylated proteins. (**A**) Subcellular distribution of up-regulated and down-regulated differentially acetylated proteins. (**B**) GO-CC and GO-MF analyses were performed for up-regulated and down-regulated differentially acetylated proteins. GO: Gene Ontology; CC: Cellular Component; MF: Molecular Function.

**Figure 6 biology-13-00559-f006:**
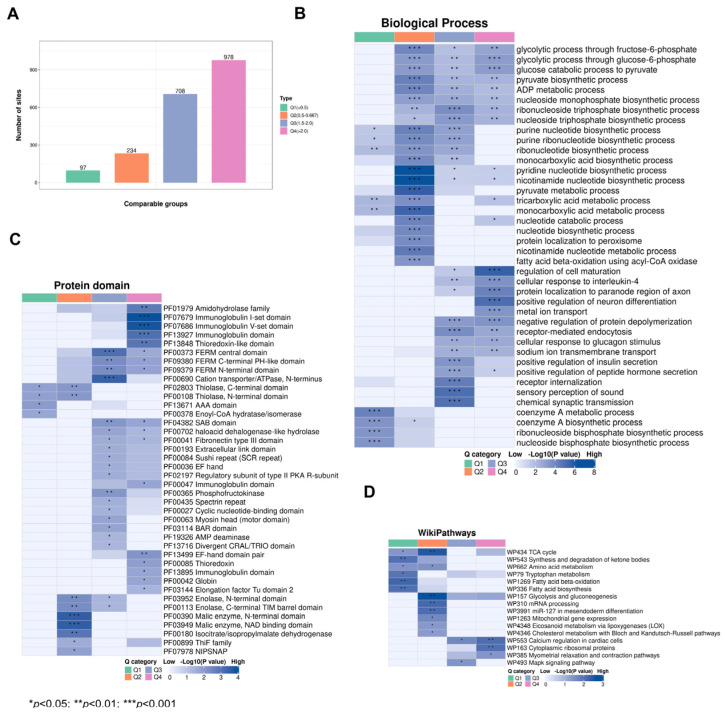
Functional analysis of acetylated proteins with foldchange. (**A**) Differential acetylation sites were divided into four groups according to foldchange; (**B**) WikiPathways analysis, (**C**) protein domain analysis, and (**D**) GO-BP analysis were performed for acetylated proteins with foldchange. GO: Gene Ontology; BP: Biological Process.

**Figure 7 biology-13-00559-f007:**
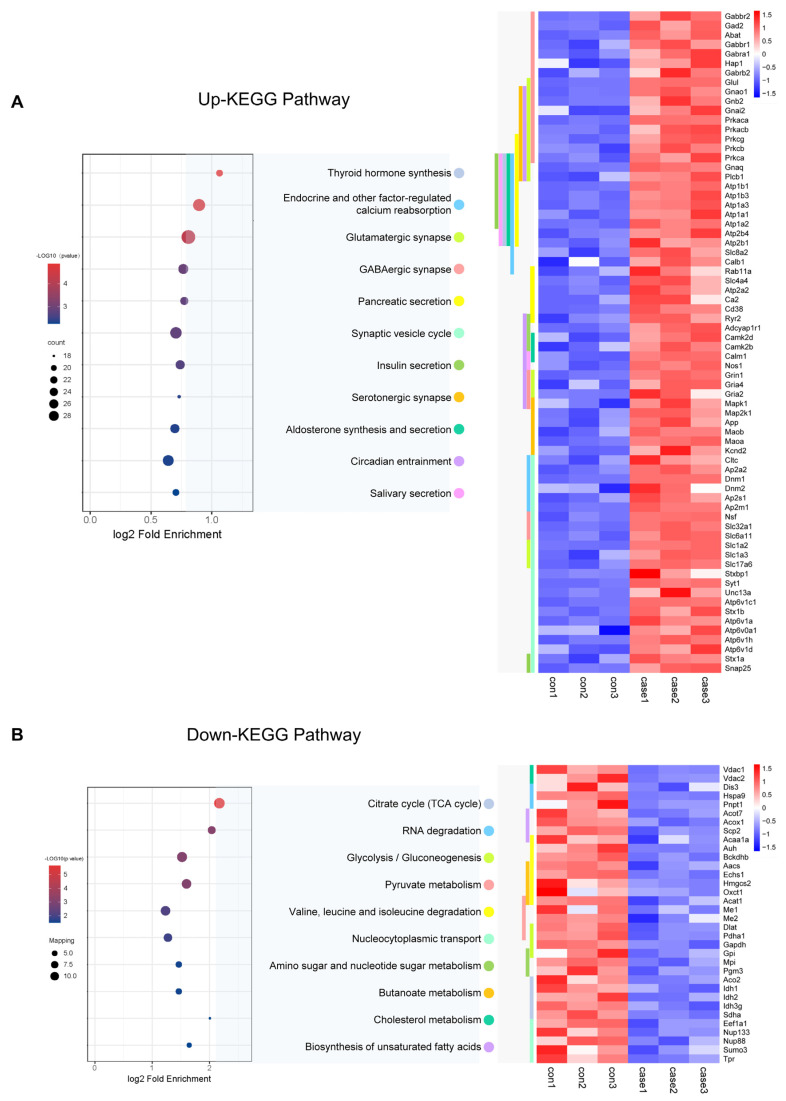
KEGG analysis of acetylated proteins. (**A**,**B**) KEGG analysis was performed for up-regulated and down-regulated acetylated proteins, and a heatmap was used to show the expression changes of acetylated proteins of important signalling pathway proteins. KEGG: Kyoto Encyclopedia of Genes and Genomes.

## Data Availability

The data presented in this study are openly available in Proteomics Identifications Database at https://www.ebi.ac.uk/pride/archive/projects/PXD051767/private (accessed on 26 April 2024), reference number PXD051767.

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
