# Peer review of "Acetylome Analyses Provide New Insights into the Effect of Chronic Intermittent Hypoxia on Hypothalamus-Dependent Endocrine Metabolism Impairment"

_biology, 2024, doi:10.3390/biology13080559_

Round 1

Reviewer 1 Report

Comments and Suggestions for Authors

Ladies and Gentlemen
I am very pleased to have been chosen as a reviewer of such an important publication.
It is extremely important to obtain information about changes in metabolism in patients with symptoms of hypoxia caused by OSA.
The results of the work can be used for further work and everyday clinical practice.

The work is structured in a typical way, it consists of an extensive, clearly presented and substantive introduction, the objectives of the study are clearly stated, the method and material are very well presented, the results of the work are extremely interesting, and the conclusions correspond to the presented objectives of the work.

After minor stylistic corrections, I recommend accepting the job without additional delays.

Author Response

Many thanks for your kind support regarding the novelty of our manuscript and the careful analysis on the conceptual significance. We made a further improved the manuscript.

Reviewer 2 Report

Comments and Suggestions for Authors

The manuscript "Acetylome analyses provide new insights into the effect of chronic intermittent hypoxia on hypothalamus-dependent endocrine metabolism impairment" reports the results of a combination of two experiments, one of them is a proteomics analysis of acetylome in a mouse hypoxic model, and the other one is a metabolomics analysis of plasma from five OSA patients.

Unfortunately, the current text lacks many important details and includes serious flaws, which require re-evaluation of the data.

First of all, the authors should pay much more attention to the figures captions, because current variants are too scarce and sometimes the logic of their combinations should be explained (e.g. in figure 1). In my opinion, the figure should be divided into two parts, one of which should be added as a general experimental pipeline in methods.

The methods and figures should include not only full details of the procedures, but also include the number of samples/animals/patients. Currently these details are missing practically everywhere. The chapter on patients should also include the details on control individuals, which are e.g. not included in the table S16.

The authors should clarify, how their current work differs from their recent paper (https://doi.org/10.3389/fnmol.2024.1324458) which is only cited in methods. Depending on the clarification, the current paper, having 15% of text in common with the previous one, is either an attempt to re-use/re-publish the (part of?) previous data or a completely different study. In any case, the previous results should be carefully mentioned, and the differences and similarities of the results should be clarified.

As for the metabolomics part, its results likely contain misleading data such as the presence of drugs and plant/fungi metabolites in the blood of children. First of all, the description of metabolites should be improved, because current version even does not explain abbreviations. Secondly, an explanation is needed, how did it happen that compounds such as allopurinol, edetic acid, licarin C and many others has been identified in the blood of children. It's highly unlikely, these drugs were taken by any of them, so the data analysis pipeline should be corrected for a mistake.

Only after such a correction further global analysis and discussion can be done.

Author Response

Many thanks for your commendation on the manuscript and very valuable criticisms on our weakness. We absolutely accepted your advice and made revisions on the manuscript.

Question 1 (Q1):  First of all, the authors should pay much more attention to the figures captions, because current variants are too scarce and sometimes the logic of their combinations should be explained (e.g. in figure 1). In my opinion, the figure should be divided into two parts, one of which should be added as a general experimental pipeline in methods.

Response: We greatly appreciate your advice. As a result of your inspiration, we have made adjustments to the order of the graphs for improving combinational logic. The flow chart depicting protein acetylation analysis in the mouse hypothalamus, originally featured in Figure 1B, has been included as a graphic summary in the pediatric Plasma metabolomics section. Additionally, HE staining of the hypothalamus, previously displayed in Figure 1A, has been relocated to Figure 2A alongside immunohistochemistry images to illustrate both the morphology of the hypothalamus and the changes in acetylase expression.

Line 286-289

Subsequently, histological analysis utilizing HE staining demonstrated that the hypothalamic cells were dispersed and that the nuclear macrocytoplasm was deeply stained, which identified damage within the hypothalamic, including cytolysis and cytoplasmic vacuolation (Figure 2A).

Q2: The methods and figures should include not only full details of the procedures, but also include the number of samples/animals/patients. Currently these details are missing practically everywhere. The chapter on patients should also include the details on control individuals, which are e.g. not included in the table S16.

Response: According to your request, we have added the details of all the graphics. Children with OSA with an obstructive apnea-hypopnea index (OAHI) >1, as determined by polysomnograph (PSG), were enrolled in the study. Table S16 mainly demonstrated OSA children with PSG results, including the OAI, OAHI, SpO2 scheme, etc. Healthy volunteers were also recruited; however, they did not undergo PSG testing. Therefore, control information is not presented in Table S16. In addition, we do not plan to publish plasma metabolomics data from children with OSA at this time because as it is still ongoing. The purpose of this manuscript is solely to illustrate the metabolic implications of altered acetylation in the hypothalamus of CIH mice and to demonstrate that these studies have indeed been conducted to the reviewers satisfaction. In response to another reviewer's suggestion, we have removed this portion from the results section and relocated it to the discussion section.

Line 273-283

Figure 1. Proteomic changes in the hypothalamus of CIH mice. Mice in the CIH group were exposed to CIH for 3 weeks. At the end of the third week, control and CIH mice were sacrificed and the hypothalamus was preserved. Each group of three hypothala-mus is used for proteomics and acetylomics. (A) PCC analysis was used to compare the intensity values of all samples; (B) The total number of peptides and proteins that were identified after data filtering; (C) Volcano plot and (D) hierarchical clustering heatmap present DEPs, which were screened according to foldchange >1.5 or <0.667, with P value <0.5; (E) COG, (F) GO-CC, (G) GO-MF, and (H) KEGG analyses were performed on DEPs; (I) Heatmap shows the protein ex-pression in endocrine-related pathways.

CIH: chronic intermittent hypoxia; PCC: Pearson correlation coefficient. DEPs: differentially ex-pressed proteins; COG: Clusters of Orthologous Groups; GO: Gene Ontology; CC: Cellular Com-ponent; MF: Molecular Function; KEGG: Kyoto Encyclopedia of Genes and Genomes.

Line 320-333

Figure 2. Expression changes of acetylation-related enzymes in the hypothalamus of CIH mice. (A) HE staining. (B) - (C) IHC staining was used to detect the protein expression of ACSS2, ACADS, ACOX1, and Kac. All experiments were repeated three times. All data are presented as the mean ± standard deviation.

IHC: Immunohistochemistry; ACSS2: acetyl-CoA synthetase; ACADS: acyl-CoA dehydrogenase short chain; ACOX1: acyl-CoA oxidase; Kac: acetylated-Lysine.

Figure 3. Characteristic motif analysis of acetylation sites in the hypothalamus of CIH mice. (A) PCA was used to characterize the signatures of the control and CIH groups and distinguish them; (B) Numbers of identified and quantified proteins and Kac sites; (C) Hierarchical clustering heatmap and (D) volcano plot present differentially Kac sites, which were screened according to foldchange >1.5 or <0.667, with P value <0.5; (E) The proportion of Kac proteins to total identified proteins; (F) The number of Kac sites in Kac proteins.

PCA: principal component analysis; Kac: Lysine acetylation

Line 354-359

Figure 4. Analysis of molecular characteristic groups of acetylation sites. (A) IceLogo method shows the frequency changes of amino acids near acetylation sites; (B) Heatmap showed the structural preference of all acetylation sites; (C) Statistics of motif changes at acetylation sites; (D) Statistics of the α-helix, β-strand, and coil of the identified Kac sites. (E) Statistics of the surface accessibility of the identified Kac sites.

Kac: Lysine acetylation.

Line 385-389

Figure 5. Subcellular distribution and GO analysis of acetylated proteins. (A) Subcellular dis-tribution of up-regulated and down-regulated differentially acetylated proteins. (B) GO-CC and GO-MF analyses were performed for up-regulated and down-regulated differentially acetylated proteins.

GO: Gene Ontology; CC: Cellular Component; MF: Molecular Function.

Line 414-417

Figure 6. Functional analysis of acetylated proteins with foldchange. (A) Differential acetylation sites were divided into four groups according to foldchange; (B) Wiki pathway analysis, (C) pro-tein domain analysis, and (D) GO-BP analysis was performed for acetylated proteins with fold-change.

GO: Gene Ontology; BP: Biological Process.

Line 462-465

Figure 7. KEGG analysis of acetylated proteins. (A)-(B) KEGG analysis was performed for up-regulated and down-regulated acetylated proteins, and heatmap was used to show the ex-pression changes of acetylated proteins of important signaling pathway proteins.

KEGG: Kyoto Encyclopedia of Genes and Genomes.

Q3: The authors should clarify, how their current work differs from their recent paper (https://doi.org/10.3389/fnmol.2024.1324458) which is only cited in methods. Depending on the clarification, the current paper, having 15% of text in common with the previous one, is either an attempt to re-use/re-publish the (part of?) previous data or a completely different study. In any case, the previous results should be carefully mentioned, and the differences and similarities of the results should be clarified.

Response: Our recent paper describes the effects of altered hippocampal protein acetylation on cognitive function in CIH mice. In recent paper, we utilized 6-week-old mice CIH for a duration of 4 weeks to established an OSA model. In addition, our ongoing project focusing on plasma metabolomics changes in OSA children has revealed initial evidences of metabolic abnormalities in OSA children patient population. Building upon these findings and considering the reported the link between Kac- endocrine metabolism-hypothalamus-OSA (as we described in the introduction section), we sought to investigate whether CIH induces endocrine metabolic abnormalities through protein acetylation changes in the hypothalamus. To better mimic pediatric patients, we used 3-week-old mice and the shortened corresponding CIH duration was shortened to 3 weeks for the present study. While our recent papers serve as the primary research foundation for this manuscript, it is important to note that they represent distinct studies with no overlapping data.  We regret not providing a detail discuss regarding their connection initially; however, we have addressed this issue in the revised discussion section. The repetition between them primarily exists within the methods and results section due to their shared  use of the protein and acetylated omics techniques. We apologize for any duplications and have taken steps to minimized their occurrence.

Q4: As for the metabolomics part, its results likely contain misleading data such as the presence of drugs and plant/fungi metabolites in the blood of children. First of all, the description of metabolites should be improved, because current version even does not explain abbreviations. Secondly, an explanation is needed, how did it happen that compounds such as allopurinol, edetic acid, licarin C and many others has been identified in the blood of children. It's highly unlikely, these drugs were taken by any of them, so the data analysis pipeline should be corrected for a mistake.

Only after such a correction further global analysis and discussion can be done.

Response: Thank you very much for your advice! We would like to clarify that we do not currently intend to publish data on plasma metabolomics in children with OSA, as collection, detection, and analysis of all human samples have not yet been concluded. The relevant results have been incorporated into the discussion as per your suggestion, including the full names of all metabolites in Table S16. Upon careful review, it has come to our attention that Edetic Acid may have originated from improper handling of EDTA anticoagulant tubes used for blood collection. thorough consideration, we have decided to remove it from the identified differential metabolites. Given the common use of some Traditional Chinese medicine and patent Chinese medicines in China, we suspect that they may be the source of some metabolites. For example, Licarins may be derived from Myristicae Semen or Magnoliae Flos and are used for anti-viral, anti-bacterial or anti-inflammatory purposes [1]. As a result, we believe it is necessary to avoid the intake of Traditional Chinese medicine in future plasma collection. Alternatively, some metabolites may be associated with the intake of certain foods, such as allopurinol [2],  therefore these will remain included in the list for now. Despite literature support for their in pediatric plasma remains to inconclusive at this time; hence this part of the data will not be published alongside this manuscript. Fortunately, however after carefully re-evaluating functional enrichment analysis results regarding differential metabolites identified above did not impact important signaling pathways' enrichment outcomes. Therefore, discussion pertaining to altered metabolic function in children with OSA and altered hypothalamic acetylation in CIH mice will remain intact.  Following an expansion of sample size in future studies; complete metabolomics data will indeed be made available for publication.

References:

  1. Alvarenga DJ, de Figueiredo Peloso E, Marques MJ, de Souza TB, Hawkes JA, Carvalho DT: Natural and Semi-synthetic Licarins: Neolignans with Multi-functional Biological Properties. Revista Brasileira de Farmacognosia 2021, 31(3):257-271.
  2. Zhang J, Du Y, Sun Y, Zhou L, Xu J, Sun J, Qiu T: Effect of orange solid waste diet on flesh quality and metabolic profile of common carp (Cyprinus carpio). Food chemistry 2023, 425:136427.

Reviewer 3 Report

Comments and Suggestions for Authors

In this study, the authors conducted acetylome, proteomic and metabolic analysis for infantile mice hypothalamus to examine the effects of pediatric CIH-induced global protein acetylation on hypothalamic function and endocrine metabolism. They found chronic intermittent hypoxia induced significant alterations in acetylated proteins which exhibited disruptions primarily in hypothalamus-dependent endocrine metabolism. The study is novel and promising, and paper was well written. However, apparent figure reuse in different IHC staining was found in Figure 3, which could be either unconscious mistake or obvious data manipulation. Therefore, the authors should provide convincing explanation.

1. In Figure 3, control images for ACADS and ACOX1 IHC staining are exactly the same, the authors should explain why.  

2. Some sentences are confusing and redundant, like lines 78-79: “whether CIH alters the function of the hypothalamus in other brain regions”, lines 81-82: “we characterized hypothalamus global Kac in 3-week CIH-induced wild-type C57B/L6J mice for 3 weeks”, lines 211-212 “To investigate the impact of CIH on behavioural changes and Kac in the hypothalamus”, do you have any behavioral data to support your statement?

3. CIH group in the figures should be marked as “CIH”, instead of “IH”, which will be consistent with the main text and avoid confusion.

4. Since the clinic data in Result 3.7 (line 441) were not shown, I suggest moving this part to Discussion section for additional discussion.

Comments on the Quality of English Language

Overall, the writing of this paper is excellent, except a few parts which sound confusing and redundant, and I have pointed out above.

Author Response

Many thanks for your commendation on the manuscript and very valuable criticisms on our weakness. We absolutely accepted your advice and made revisions on the manuscript.

Question 1 (Q1): In Figure 3, control images for ACADS and ACOX1 IHC staining are exactly the same, the authors should explain why.  

Response: We sincerely apologize for the inadvertent uploaded of incorrect figure! Following a thorough review, we have rectified all the erroneous data and images.

Q2: Some sentences are confusing and redundant, like lines 78-79: “whether CIH alters the function of the hypothalamus in other brain regions”, lines 81-82: “we characterized hypothalamus global Kac in 3-week CIH-induced wild-type C57B/L6J mice for 3 weeks”, lines 211-212 “To investigate the impact of CIH on behavioural changes and Kac in the hypothalamus”, do you have any behavioral data to support your statement?

Response: Thank you very much for your correction, which is greatly beneficial for us to improve the English in the manuscript.

Line 90-95

However, whether CIH alters the function of other brain regions such as the hypothalamus through Kac mechanisms is unknown.

In the present study, 3-week-old mice were subjected to CIH for 3 weeks to mimic children with OSA before adulthood. The CIH-induced global Kac changes in the hypothalamus and the possible functional abnormalities were then investigated.

Line 225-227 (we deleted behavioral)

To investigate the impact of CIH on Kac changes in the hypothalamus, 3-week-old male C57BL/6J mice were subjected to a hyperbaric oxygen chamber for a period of 3 weeks to simulate CIH as they transitioned from adolescence to adulthood.

Q3: CIH group in the figures should be marked as “CIH”, instead of “IH”, which will be consistent with the main text and avoid confusion.

Response: Thank you for your correction. We have revised in all figures according to your suggestion.

Q4: Since the clinic data in Result 3.7 (line 441) were not shown, I suggest moving this part to Discussion section for additional discussion.

Response: Thanks for your advice! We are completely agreed with what your suggestions. We have removed mentioned part and included additional discussion in our paper.

Round 2

Reviewer 2 Report

Comments and Suggestions for Authors

Dear authors,

Thank you for your reply and the revision of several raised points (Q1-Q3).

Focusing on the main question, there is a complex situation with the metabolomics data within this manuscript. As you write in the response, you "do not plan to publish plasma metabolomics data from children with OSA at this time because as it is still ongoing."  But in the current version you do try to publish the data, even though they are not full data.
Because of the low reliability of the currently presented metabolomics results and you "not planning" to publish these data, I suggest to exclude these data and all its mentions from the text. I have to insist on that, because of the example of
edetic acid (mentioned in my original Q4) and the other ones. The drugs, which you describe as potentially coming from traditional medicine, or other drugs should at least be added with careful description of medical history of patients. Alternatively, one may ask to support your data with purchased samples containing heavy isotopes... Currently, only a few mentioned examples raise questions on how these drugs could be detected in the blood of children. If you further insist on adding these data to the paper, we'll have to go through the table row by row and explain the potential origin of the drugs in each patient (likely as additional columns of the supplementary table or somehow else).

You also wrote, that "The purpose of this manuscript is solely to illustrate the metabolic implications of altered acetylation in the hypothalamus of CIH mice and to demonstrate that these studies have indeed been conducted to the reviewers satisfaction" (the mentioned reviewer is another one, as I can understand).
To my mind, the implications of altered acetylation
in the hypothalamus of CIH mice are difficult to combine with the metabolomics of children, because these are two different systems/models. You should either evaluate the metabolites in your mouse model or add a study of the acetylome of patients (not possible). Thus you should not try mixing these two completely different systems, but rather focus on the mouse model, which is described relatively well.
To satisfy another reviewer, you might just add the data as NON-published material for him/her...

Of course, the suggested removal of the currently questionable matabolomics data from the manuscript requires extensive editing of the text, but it's worth doing. I expect, a manuscript with more focus on the MS data and the animal model would have even better logic than the current version combining two different pieces of data from completely different models.

Author Response

Thank you very much for your advice. Following your suggestion, we have removed all pediatric plasma metabolomics data and discussions to ensure the reliability and logic of the manuscript.